# Detection and Characterization of the Eukaryotic Vacant Ribosome

**DOI:** 10.3390/ijms27010308

**Published:** 2025-12-27

**Authors:** Colin E. Delaney, Attila Becskei

**Affiliations:** Biozentrum, University of Basel, Spitalstrasse 41, 4056 Basel, Switzerland; colin.delaney@unibas.ch

**Keywords:** starvation, 80S ribosome, monosome, polysome, *Saccharomyces cerevisiae*, neuron, heat-shock, Stm1, Lso2, YfiA, SERBP1, CCDC124

## Abstract

Upon transcription, most mRNAs associate with the small ribosomal subunit, after which a fully translating ribosome assembles. Under starvation or stress, however, mRNA–ribosome associations are blocked and many mRNAs are instead sequestered with specific RNA-binding proteins into stress granules or other subcellular condensates, a process that has been extensively studied. In contrast, much less attention has been paid to the fate of ribosomes under these same conditions. Ribosomes can remain fully assembled but unbound to mRNA, entering an inactive, dormant state. Dormancy is often supported by specific protein factors which protect ribosomes from degradation and facilitate reactivation once growth conditions improve. In this review, we highlight that dormant ribosome states are well defined in prokaryotes, in part because they possess distinct and experimentally tractable features, such as stable vacant 100S dimers. In eukaryotes, by contrast, analogous disomes are largely absent, making their discovery more indirect and method-dependent. We therefore focus on how evidence for eukaryotic dormant ribosomes has been assembled through multiple independent findings and how their interpretation depends critically on the experimental approaches used to study them. Finally, we consider atypical ribosomal states, such as translationally inactive polysomes in neurons, which underscore the context-dependent nature of ribosome activity.

## 1. Introduction

The fundamental sequence of events in eukaryotic translation has been known for decades: an mRNA first associates with the small (40S) ribosomal subunit, the large (60S) subunit then joins to form an 80S ribosome, and one, two or more ribosomes translate the mRNA concurrently. This mechanistic framework defines several principal molecular states, including free mRNA, free 40S and 60S subunits, 40S-bound mRNA, and actively translating ribosomes in the form of monosomes, disomes, and polysomes. The balance between the various ribosomal subunits and assembly states is tightly regulated [1]. Polysome profiling, which separates these complexes by sedimentation through a sucrose gradient, remains a key method for resolving and quantifying these states (Figure 1).

In addition to these canonical translating species, non-translating ribosomal complexes have also been described, most notably vacant monosomes and disomes. These complexes are less well-characterized and consist of fully assembled ribosomes lacking bound mRNA and typically arise under starvation or stress. They appear in the literature under several partially overlapping terms, including vacant, idle, empty, inactive, and silent ribosomes [2]. For the purposes of this review, we will treat these terms as interchangeable.

When cells enter stationary phase, growth slows or ceases and the demand for protein synthesis drops. In response, bacteria reduce ribosome abundance through active degradation [3], while eukaryotes rely on multiple turnover pathways, most prominently selective autophagy of ribosomes (ribophagy) [4]. Not all ribosomes are degraded, however; a fraction enters a dormant state in which the small and large subunits remain assembled as an 80S ribosome but lack mRNA and translational activity. These hibernating ribosomes were first described in prokaryotes [5], and for a long time, ribosome hibernation was believed to be restricted to bacteria and plant plastids [6]. Systematic exploration of potential eukaryotic counterparts began only much later.

In this review, we examine the factors that contributed to this temporal gap, focusing on the methodological and mechanistic differences that may have obscured or delayed the recognition of silent ribosomes in eukaryotes. We then summarize recent advances demonstrating that dormant or silent ribosomes occur in organisms ranging from yeast to mammals. We begin by outlining key principles from bacterial hibernation—drawing only on concepts necessary for comparison—before turning to eukaryotic systems. We focus specifically on budding yeast and mammalian cells, and paying particular attention to neurons, where monosomes and polysomes can assume atypical roles.

**Figure 1 ijms-27-00308-f001:**
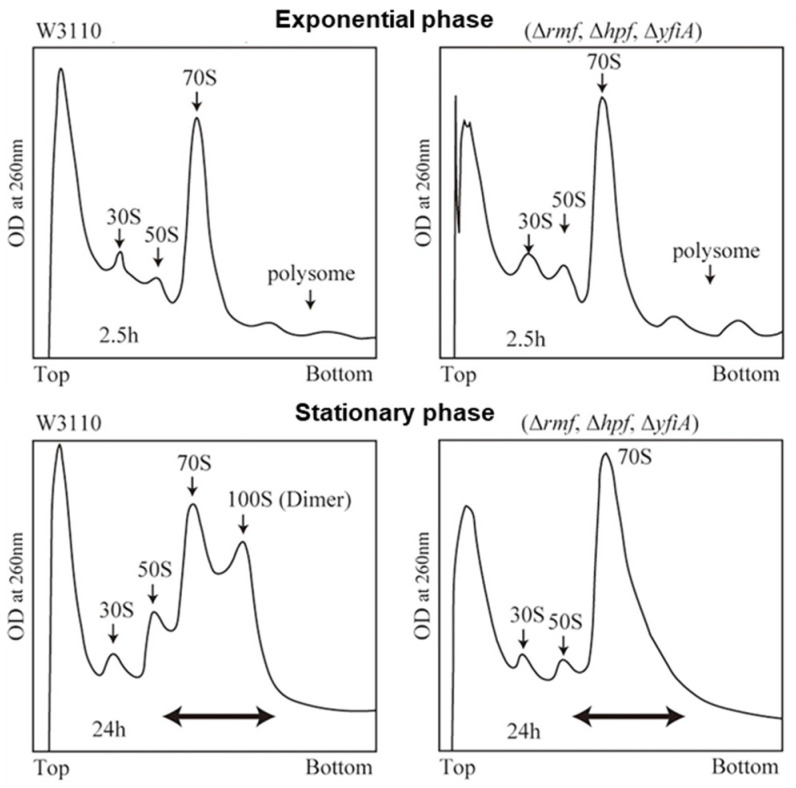
Polysome profiles of *E. coli* cells in exponential growth (**top**) and stationary phases (**bottom**) in wild-type cells (W3110, **left**) and cells in which the genes of the hibernation factors are deleted (Δ*rmf*, Δ*hpf* and Δ*yfiA*, **right**). Adapted from Yoshida et al. [7] in a modified version. Originally published in *Frontiers in Molecular Biosciences* under a CC-BY license.

## 2. Dormant Ribosomes in Prokaryotes and Plant Chloroplasts

In bacteria, one of the most striking alterations of the translational apparatus during entry into stationary phase is the progressive increase in the 100S dimer peak, accompanied by a concomitant decrease of 70S monosomes and polysomes [8] (Figure 1). Although early biochemical studies hinted that 100S ribosomes represented a dormant yet rapidly reactivatable state, the physiological relevance of 100S dimers was not firmly established until the 1990s, in part because research had predominantly focused on exponentially growing cells [5,9]. In *E. coli* and many other bacteria, the formation of the 100S ribosomes is mediated by ribosome modulation factor (RMF) and hibernation promoting factor (HPF). Gram-positive bacteria lack RMF but encode a long HPF isoform (lHPF) whose extended C-terminal region promotes dimerization by direct protein–protein interactions [10].

In addition to 100S dimers, bacteria also produce hibernating monosomes mediated by the ribosome-associated inhibitor A (RaiA), also known as YfiA (Protein Y) in some species (Figure 2). RaiA is homologous to HPF but carries a short C-terminal extension that likely interferes with RMF binding and therefore disfavors 100S dimerization. RaiA overexpression increases the proportion of intact 70S ribosomes relative to dissociated subunits during stationary phase, consistent with the hypothesis that RaiA stabilizes monosomes and thereby shortens the lag phase upon growth resumption [11].

Recent work demonstrated that the three *E. coli* hibernation factors RMF, HPF, and RaiA act cooperatively to confer ribosome protection during carbon starvation. Cells lacking all three show severely impaired regrowth and accumulate 70S ribosomes with fragmented 16S rRNA, whereas rRNA in wild-type 100S dimers remains intact. The fragmentation is suppressed in strains lacking RNases YbeY and RNase R, suggesting that the hibernation factors protect ribosomes by physically blocking ribonuclease access [6].

A related mechanism operates in plant chloroplasts, evolutionarily derived from bacteria. Chloroplasts encode the plastid-specific ribosomal protein 1 (PSRP1), the chloroplast ortholog of bacterial HPF. PSRP1 dimerizes, a prerequisite for inducing ribosome dimerization. Interestingly, PSRP1 induces 100S-like dimers with purified bacterial ribosomes in vitro, but not with purified chloroplast ribosomes. Furthermore, endogenous chloroplast 100S particles have not been detected; nevertheless, PSRP1 protects ribosomes by preventing chloroplast rRNA degradation in the dark. Together, these findings suggest that although PSRP1 retains prokaryotic features, chloroplast ribosome protection no longer relies on dimerization-based hibernation but instead reflects lineage-specific adaptations tailored to their physiological environment [12].

## 3. From Bacteria to Eukaryotes: Challenges to Identifying Vacant Ribosomes

In prokaryotes, two key features have enabled a systematic and detailed characterization of dimeric hibernating ribosomes: (i) distinctive polysome profile signatures and (ii) well-defined, stress-regulated hibernation factors. As a result, by the early 2000s, a nearly complete picture of bacterial ribosome hibernation had emerged.

The dimeric form is particularly well described, as it generates a characteristic disome peak in stationary-phase polysome profiles (Figure 1). Its molecular basis is firmly established: deletion of *rmf* and *hpf* abolishes this peak, demonstrating their essential role in dimer formation [8]. Moreover, *rmf* transcription negatively correlates with growth rate and is induced by diverse stresses, including amino-acid starvation, heat and cold shock, ethanol, pH and osmotic stress, and envelope perturbation [5].

In contrast, dimeric silent ribosomes are largely absent in eukaryotes—with only isolated reports discussed below [13,14]—potentially explaining why the characterization of eukaryotic silent ribosomes has progressed more slowly. In eukaryotic cells, silent ribosomes are generally monomeric. Yet even for these monomeric forms, eukaryotes lack the two key features that greatly facilitated the study of their prokaryotic counterparts (Table 1, [15]).

First, the monomeric hibernation factor in bacteria, RaiA, is homologous to the dimer-associated factor RMF, providing an evolutionary and mechanistic link that guided experimental interpretation. In addition, RaiA overexpression produces a clear and quantifiable increase in the monosome peak at the expense of the subunits [11], providing a clear experimental signature of its function.

Second, bacterial hibernation factors show strong stress-dependent induction, such as the marked upregulation of YfiA (a RaiA homolog) in stationary phase [16], making them easy to identify as stress-responsive ribosome regulators. By contrast, eukaryotic candidates such as Stm1 are already present during exponential growth and do not increase under starvation or stress [17], rendering their functional role less immediately apparent. Nevertheless, important gaps remain even in prokaryotes. For instance, a pronounced monosome peak emerges in *E. coli* under carbon and nitrogen starvation, yet the deletion of *raiA* has no detectable effect on polysome profiles [18], indicating that the underlying mechanisms remain to be elucidated.

Shifts from polysome to monosome dominance in eukaryotes occurs under diverse stresses—including starvation, cold shock, heat shock, and lithium exposure—and were historically interpreted as a result of inhibition of initiation or early elongation rather than formation of vacant ribosomes [19,20]. These two mechanisms are experimentally challenging to distinguish (Figure 3).

For example, during heat stress, ribosomes stall shortly after initiation. A detailed analysis in mammalian cells revealed a response highly similar to that induced by harringtonine, which blocks elongation during the first peptide bonds [21]. After harringtonine treatment, cells accumulate monosomes and light polysomes at the expense of heavy polysomes, a pattern similar to the one observed after heat shock [22]. Ribosome profiling showed ribosome accumulation close to the start codon that have led to the following conclusion “mRNAs with ribosomes paused at typical locations (around 200 nt) are expected to migrate mostly as lighter polysomes or monosomes, depending on whether additional ribosomes accumulate upstream of the pause.” [22]. This early elongation block is augmented when heat shock proteins are inhibited but alleviated by HSP overexpression, demonstrating their direct role in regulating translation elongation near the start codon. Similar HSP-dependent pausing of ribosomes has also been observed in proteotoxic stress [23]. A related mechanism exists under lithium stress, where inhibition of translation initiation in galactose-grown yeast cells can be alleviated by eIF4A overexpression, suggesting interference with early initiation steps rather than elongation [20]. Several additional mechanisms link cellular stress responses to translational elongation, exemplified by stress-induced covalent modifications of the tRNA anticodon stem–loop. These modifications, particularly at wobble positions, can modulate codon–anticodon pairing efficiency and thereby alter elongation dynamics [24]. For example, loss of cytosine methylation in a specific tRNA causes ribosome stalling, reduces translation efficiency, and ultimately impairs *C. elegans*’ ability to adapt to elevated temperatures [25].

Recent results indicate, however, that at least some stresses—such as heat shock—primarily act through the formation of silent ribosomes [15]. Why, then, have these phenomena been interpreted as initiation/elongation blocks rather than ribosome dormancy? A key factor is that early elongation is intrinsically rate-limiting in eukaryotes, which show a much slower transition from initiation to elongation than *E. coli*. This slowdown is linked to the prolonged residence time of eIF5B on the 80S ribosome after subunit joining [26,27].

Taken together, these findings indicate that increases in monosomal (and lighter polysomal) fractions and the concomitant loss of heavy polysomes in eukaryotes may arise from either formation of silent ribosomes or early-elongation/initiation stalls (Figure 3), and that these processes may co-occur. Given that alternative mechanisms that can lead to the accumulation of monosomes—particularly elongation-blocked versus vacant ribosomes—it is unsurprising that the identification of eukaryotic silent ribosomes lagged by ~20 years behind that in *E. coli*.

### 3.1. Protein Interactions with Dormant Ribosomes in Yeast

The identification of ribosome preservation mechanisms in budding yeast *Saccharomyces cerevisiae* began with the characterization of Stm1, a protein that binds independently of mRNA to ribosomes and supports cell viability during prolonged starvation [28]. Structural studies of ribosomes purified from cells subjected to brief glucose starvation showed that Stm1 is associated with the 40S subunit, where it blocks the mRNA entry channel [29]. Owing to its translation-inhibiting effect (Table 1), Stm1 was proposed to preserve ribosomes in an inactive state during nutrient limitation, thereby serving as a functional analog of the stress-induced ribosome preservation factors found in bacteria and chloroplasts [29].

**Table 1 ijms-27-00308-t001:** Key studies contributing to the characterization of Stm1-bound vacant ribosomes in yeast.

Year	Authors (First/Last)	Finding	Condition
2006	Van Dyke/Van Dyke [28]	Stm1 associates with ribosomes independently of mRNA; *stm1*Δ cells show reduced viability under prolonged nitrogen starvation.	Long-term nitrogen starvation (2 to 6 days).
2011	Balagopal/Parker [30]	Stm1 overexpression causes stronger growth inhibition in *dom34*Δ strains than in wild type, suggesting that Stm1 stalls ribosomes in vivo and that Dom34/Hbs1 releases Stm1-stalled ribosomes.	Growth on rich medium under mild cold shock (16 °C).
2011	Ben-Shem/Yusupov [29]	Crystal structure reveals only one non-ribosomal protein, Stm1. Stm1 is suggested to clamp the two subunits preventing their dissociation, and inhibiting translation by excluding mRNA binding.	Short-term glucose starvation (30 °C, 10 min).
2013	Van Dyke/Van Dyke [17]	Ribosomal protein levels are similar in wild-type and *stm1*Δ cells after one day in quiescence but diverge after four days; Stm1 overexpression prevents ribosome degradation.	Long-term starvation: stationary phase (4 days).
2014	van den Elzen/Séraphin [31]	Stm1-bound 80S ribosomes are substrates for Dom34/Hbs1/Rli1-mediated subunit splitting in vitro. Deletion of *STM1* suppresses the requirement for Dom34-Hbs1 to restart translation in vivo.	Short-term glucose starvation combined with mild cold shock (10 min, 16 °C).

Subsequent experiments, however, showed that Stm1 is not required for the rapid formation of silent ribosomes. Deletion of *STM1* did not eliminate the prominent monosome peak induced by short glucose depletion (≈10 min) [31]. Similarly, silent ribosomes also accumulate after heat shock (≈30 min), yet *stm1*Δ strains again showed no reduction in monosome accumulation [15]. Thus, the formation of vacant ribosomes during short-term stress (<1 h) occurs independently of Stm1.

By contrast, Stm1 becomes important during prolonged starvation and quiescence. During long-term starvation, Stm1 contributes to ribosome preservation [28]. In 4-day stationary cultures, *STM1* deletion reduces, whereas *STM1* overexpression increases, the monosome peak [17]. In the presence of Stm1, protein synthesis is resumed faster after exit from quiescence; polysome reassembly is impaired in *stm1*Δ cells, indicating that Stm1 primarily preserves ribosomes for efficient reactivation, rather than being required for silent ribosome formation itself [17].

Together, these findings support the following framework: acute stress triggers the rapid formation of vacant ribosomes independently of Stm1, whereas Stm1 preserves ribosomes during extended starvation, thereby enabling rapid translation restart—a process further supported by Dom34-mediated reactivation (Table 1). Thus, vacant ribosomes associated with Stm1 can be detected in both short-term and long-term starvation or stress; however, the functional role of Stm1 can be assigned reliably only in long-term experiments due to the slow degradation of the ribosomes.

Following the stepwise characterization and identification of Stm1-bound vacant ribosomes, a systematic study examined which proteins increase their association with ribosomes upon glucose deprivation ([32], Table 2). Of the proteins enriched under these conditions, the late-annotated small open reading frame protein Lso2 was characterized in detail. Lso2, a short 92-aa protein, binds to the ribosomal GTPase activation center, a critical hub that governs accurate progression through all substeps of translation. Notably, this binding mode is conserved in the human ortholog of Lso2, coiled-coil domain-containing protein 124 (CCDC124), suggesting an evolutionarily conserved role in translational regulation.

**Table 2 ijms-27-00308-t002:** Key studies contributing to the identification and initial characterization of Lso2-bound vacant ribosomes.

Year	Authors (First/Last)	Finding	Condition
2018	Wang/Gilbert [32]	Lso2 shows increased association with ribosomes upon starvation; recovery from starvation is accelerated in the presence of Lso2, as indicated by earlier reappearance of polysomes.	Short term glucose starvation (2 h).
2020	Wells/Beckmann [33]	Cryo-EM analysis identifies two distinct populations of idle, translationally repressed 80S ribosomes: one bound to Stm1/SERBP1 together with eEF2, and the other bound to Lso2 (CCDC124). These populations differ in both composition and ribosome conformation.	Human embryonic kidney ([HEK]293T) cells at high confluency.

To investigate the effect of Lso2 on translation, the authors performed ribosome footprinting to quantify the abundance of empty ribosomes. These RNA-seq-based measurements of mRNA associated with ribosomes were normalized to ribosome numbers. The ribosome numbers were estimated using the average intensity of 18S and 25S rRNAs quantified from agarose gels. During recovery from glucose deprivation, ribosomes consisted predominantly of non-translating, empty monosomes. Ribosome profiling under these conditions revealed that *lso2*Δ cells accumulate both empty monosomes and translating 80S ribosomes stalled at start codons. These findings indicate that Lso2 acts at a later stage than Stm1: whereas Stm1 primarily protects ribosomes during stress, Lso2 facilitates the efficient reactivation of translation during recovery [32]. In line with this model, reconstituted Lso2–80S ribosome complexes are more efficiently split by Dom34 than 80S ribosomes alone or those bound to Stm1 [33].

A distinct but non-exclusive mechanism operates in *Schizosaccharomyces pombe* during deep quiescence (seven days of glucose depletion). Here, mitochondrial fragmentation triggers sequestration of cytosolic ribosomes on mitochondria. Cryo-EM analyses revealed that these ribosomes are devoid of mRNA and tRNA, and assemble into higher-order oligomeric arrays on the mitochondrial surface. This anchoring is mediated by the ribosomal protein Cpc2/RACK1, which binds the outer mitochondrial membrane via the small subunit [34].

Whether a similar process exists in budding yeast remains unresolved. However, it is notable that the mitochondrial protein Fmp45 is one of the three most strongly heat-induced monosome-associated proteins (Mbf1, Lso2, Fmp45) [15]. Fmp45 is required for long-term quiescence survival—its deletion reduces viability by ~100-fold after 16 days at 37 °C [35]. This raises the possibility that mitochondrial anchoring also occurs in *S. cerevisiae*.

### 3.2. Dimeric Hibernating Ribosomes in Eukaryotes?

Although an increase in monosomes is the predominant hallmark of ribosome hibernation in eukaryotes, scattered reports indicate that ribosome dimers can also form under certain conditions. For example, in the microsporidian parasite *Spraguea lophii*, ribosome dimers appear during sporulation with the assistance of the hibernation factor MDF1 [14]. Microsporidia are obligate intracellular parasites that were initially thought to represent an early-diverging eukaryotic lineage but are now classified as fungi, distantly related to yeasts [36,37,38].

Dimeric ribosomes have also been reported in metazoan cells, although in a highly restricted context. Polysome profiling revealed that rat—though not human or mouse—cells produce ribosomal dimers that closely resemble the 100S particles seen in bacteria. These ~110S peaks were detected only in amino-acid-starved C6 glioma cells, and the reason for this species- and cell-type specificity remains unclear [13]. Cryo-EM analysis confirmed that these sucrose gradient fractions indeed contained ribosome dimers. Notably, the formation of these dimers did not require new transcription or translation, and no proteinaceous hibernation factors were detected, echoing the oligomeric, higher-order ribosome assemblies observed in fission yeast during deep quiescence (see above).

Overall, these observations indicate that bona fide ribosome dimers are exceptional in eukaryotes—occurring only under highly specific conditions in both unicellular organisms and metazoans.

### 3.3. Vacant Monosomes in Mammals

The dominant translational response to starvation in mammalian cells resembles that observed in budding yeast; however, the functional role of yeast Stm1 is assumed by its mammalian homolog SERBP1 (serpine E1 mRNA-binding protein), while Lso2 is replaced by CCDC124 during the formation of vacant monosomes. Cryo-EM analyses of a human cell line have revealed that two major classes of hibernating 80S ribosomes can be found in the same cell population. One class associates with CCDC124 and contains a tRNA in the E site. The second class also harbors an E-site tRNA but, instead of CCDC124, contains SERBP1 together with eEF2 [32].

Because eukaryotic Stm1 is not transcriptionally induced under stress conditions, progress in understanding the regulation of ribosome hibernation has also lagged behind. Recent work has advanced this area by linking Stm1/SERBP1 regulation to nutrient-sensing signaling pathways. Stm1/SERBP1 is abundant under nutrient-sufficient conditions and is directly phosphorylated by TORC1. In this phosphorylated state, Stm1/SERBP1 associates only weakly with translating ribosomes. Upon TORC1 inhibition during nutrient depletion, Stm1/SERBP1 becomes dephosphorylated, stably binds empty 80S ribosomes, and promotes the formation of dormant ribosomes [39,40].

Additional mammalian proteins with related functions have been identified [2]. For instance, Pdcd4 (programmed cell death protein 4) binds both the 40S subunit and the 80S monosome upon serum or glucose deprivation in HEK293T cells as well as in vitro [41,42]. Its interaction with the 40S subunit is reminiscent of Stm1 binding [42] and through this interaction PDCD4 inhibits translation initiation and reinitiation [41]. However, definitive evidence that PDCD4 induces the formation of assembled vacant 80S ribosomes is still lacking [43]. Notably, PDCD4 functions as a tumor suppressor that restricts cell growth, invasion, and metastasis, and it is downregulated in many tumor types, coinciding with the global upregulation of translation observed in cancer cells. Beyond its role in cancer, PDCD4 also participates in non-oncogenic processes. In neurons, PDCD4 negatively regulates axonal growth and can be locally synthesized in adult axons in vivo. Its levels decrease at sites of peripheral nerve injury prior to nerve regeneration, suggesting that regulated PDCD4 expression contributes to neuronal plasticity and repair [44].

### 3.4. Neurons: Active Monosomes and Dormant Polysomes?

Current evidence indicates that vacant monosomes constitute the predominant dormant ribosomal species in eukaryotes. Monosomes—whether vacant or stalled during elongation—are translationally inactive. This raises a related question: to what extent are monosomes capable of supporting translation at all?

Early in vitro translation experiments using reticulocyte lysates and radiolabeled methionine provided a clear answer in this system. Robust incorporation of radioactivity was detected exclusively in polysome fractions, whereas the prominent monosome peak showed no signal above background [45]. Moreover, addition of vanadate, which inhibits translation initiation, induced polysome collapse and abolished radiolabel incorporation, further supporting the conclusion that monosomes in this context do not sustain detectable translation. Similar observations were later reported in certain cultured cell lines using puromycin-based nascent chain labeling, which likewise revealed negligible translation activity in monosome fractions [46]. Accordingly, monosomes have often been equated with non-translating ribosomes, and polysomes with actively translating ones [47,48].

There are indeed conditions under which monosomes clearly represent inactive ribosomes, particularly in experiments demonstrating the classical accumulation of monosomes following depletion of translation initiation factors [49]. However, the view of monosomes as generally inactive is challenged by observations in specialized cellular contexts. Monosomes are highly abundant in neuronal compartments, especially at synapses, where localized translation is critical for synaptic plasticity and memory formation [50]. In such settings, distinguishing between active and dormant monosomes becomes essential.

Recent evidence indicates that both scenarios occur, depending on tissue type and physiological state. In *Drosophila*, cryo-EM analyses of head tissue, including the brain, revealed that the overwhelming majority of monosomes carried at least one tRNA; only ~2% lacked tRNA, indicating that most ribosomes were translationally active [51]. A similarly high fraction of active ribosomes was found in embryonic tissue. Only two organs—ovaries and testes—displayed predominantly vacant monosomes. In testes, these monosomes associated with the interferon-related developmental regulator 1 (IFRD1). Importantly, the cryo-EM findings were corroborated by ionic-strength-based dissociation assays.

Earlier biochemical studies revealed this high-salt dissociation principle, showing that monosomal fractions from both normal and stressed cells contain a population of empty ribosomes—assembled 80S particles not bound to mRNA. Incubation of extracts under high-salt (increased ionic strength) conditions reversibly dissociates such vacant monosomes into their 40S and 60S subunits, whereas polysomal ribosome–mRNA complexes remain intact in high salt [31,52,53]. Polysome-derived ribosomes converted to monosomes by RNAse treatment also resisted disassembly in high salt conditions, indicating that salt-sensitive monosomes are devoid of RNA and/or other translation associated factors rather than bound to short or fragmented RNA [52,53]. Consistent with this, experiments supplementing 200 mM KCl to lysis buffers already containing 150 mM NaCl showed that most monosomes from ovaries dissociate under high ionic strength, whereas monosomes from head tissue remain largely intact (Table 3). Testis-derived 80S monosomes exhibited intermediate salt sensitivity—potentially due to stabilization by IFRD1—demonstrating that distinct types of vacant ribosomes differ in their biochemical stability [51].

**Table 3 ijms-27-00308-t003:** Methods to detect and characterize vacant ribosomes.

Method	Interpretation and Limitations	Examples
Crystallography; cryo-electron microscopy (cryo-EM)	Crystallography provides high-resolution structural information, whereas cryo-EM enables the identification of distinct ribosome populations with different compositions, factor occupancy, and tRNA binding states.	[29,51]
Effect of potassium chloride (KCl) on polysome profiles	Actively translating polysomes are largely resistant to elevated KCl concentrations, whereas vacant monosomes dissociate into subunits. However, a subset of vacant monosomes exhibits intermediate KCl resistance, limiting the specificity of this assay.	[31,51]
Ribosome-to-mRNA stoichiometry	Relative ribosome numbers per mRNA can be estimated by quantifying rRNA using gel staining, qPCR, or RNA-seq after filtering anomalously amplified rRNA species, providing a relative measure of ribosome vacancy.	[15,32]
Functional characterization	Hibernation: protection of ribosomes from degradation. Enhanced recovery of translation: ribosome splitting.	[17,32]

The observation that most monosomes in the brain are translationally active is particularly significant. Many neurons preferentially use monosomes to translate specific subsets of mRNAs, especially those encoding cell–cell adhesion proteins [50], which play important roles in the establishment and maintenance of neuronal identity [54].

Paradoxically, a substantial fraction of polysomes in neurons may be translationally inactive, revealing a distinct form of regulation. This phenomenon arises, at least in part, from the limited availability of the eukaryotic elongation factor eEF2. By inactivating a subset of polysomes, cells may ensure that the remaining active polysomes can recruit sufficient eEF2 to sustain rapid elongation [55]. Constraints on translational elongation in the brain are not limited to neurons. For example, in oligodendrocytes, hypomodification of specific tRNAs causes elongation pausing at cognate codons during translation, restricting elongation efficiency [56].

Partial polysome inactivation may serve purposes beyond global optimization of translation rates. One such role may be the transport and storage of translation-ready mRNAs. Neurons are known to harbor mRNA granules composed of stalled polysomes [48], often associated with fragile X mental retardation protein (FMRP) [57]. These mRNAs are maintained in a translationally repressed state, and in early hippocampal neuronal cultures more than 50% of nascent peptides are found associated with such stalled polysomes [58]. These ribosomal assemblies display distinct biochemical properties; for example, nascent peptides remain ribosome-associated even after puromycylation [58], which normally induces release of nascent peptides [59]. Collectively, these findings represent an alternative to the traditional view that polysomes are invariably translating and monosomes largely inactive, revealing a fundamentally different organization of translational control in the brain.

## 4. Methodological Challenges in Quantifying Ribosome Dormancy

### 4.1. Distinguishing Vacant Ribosomes from Elongation Blockade

The formation of vacant ribosomes represents an important facet of translational regulation and is tightly interconnected with other aspects of translation, particularly block in early elongation. Because both processes can produce similar polysome profiles—most notably an increase in the monosome peak—their experimental distinction has been challenging (see above).

Early studies suggested a prominent role for elongation control in eukaryotes, especially mammalian systems. Several lines of evidence, however, suggest that silent ribosome formation, rather than elongation blockade, plays the predominant role during heat stress in yeast. Although an increase in the monosome peak could in principle reflect a shift of most mRNAs from polysomes to monosomes, no such redistribution was observed at the level of individual transcripts, indicating that monosomes can accumulate without associated mRNAs [15]. Attempts to quantify vacant ribosome formation using RNA-seq-based mRNA:rRNA stoichiometry initially failed because the 5S rRNA exhibited anomalous amplification, particularly within the monosome fraction, leading to an overestimation of vacancy [15]. After excluding this rRNA, however, the resulting estimates were robust and internally consistent, and were independently validated using transcript-selective mRNA:rRNA qPCR ratios (Table 3). Together, these analyses support the conclusion that vacant ribosomes constitute the dominant ribosomal species in yeast under heat stress. Whether comparable approaches can be applied reliably to mammalian systems remains to be determined.

Because elongation blockade and silent ribosome formation can co-occur, methodologies capable of detecting both ribosome position and mRNA:rRNA stoichiometry are needed to quantitatively disentangle their contributions. One such approach is based on ribosome footprinting coupled to relative ribosome counts based on quantification of rRNA in gels [32] or by RNA-seq [60]. It remains unclear whether anomalous amplification affects quantification by RNA-seq, as RNase digestion used for footprinting also cleaves rRNA, unlike standard polysome profiling. Another strategy employing ultrafiltration combined with spike-in-based calibration has recently been developed to remove unwanted rRNA fragments and derive absolute ribosome:mRNA stoichiometries following RNase treatment [61]. Their uncalibrated data were in agreement with earlier studies showing the relative enrichment of ribosomes near the start codon under heat shock [22,23]. However, once footprints were normalized with the new spike-in-based calibration, the apparent ribosome pile-up near the start codon was no longer observed [61]. This finding challenges a dominant role for elongation blockade and underscores that the relative contributions of silent ribosome formation and elongation control remains to be determined.

Importantly, recent work in prokaryotes suggests that a strict separation between elongation arrest and ribosome hibernation is not always possible, as the two processes can be mechanistically coupled. This coupling is exemplified by the recently identified hibernation factor Balon in the cold-adapted bacterium *Psychrobacter urativorans* [62]. Unlike previously characterized prokaryotic hibernation factors, Balon can associate not only with vacant ribosomes but also with ribosomes engaged with mRNA and tRNAs, as Balon does not compete for binding within the mRNA channel.

One potential advantage of this mechanism is that ribosomes can enter a hibernation-like state rapidly, without waiting for translation elongation or termination to complete. Such an instantaneous response may be particularly important in slow-growing bacteria or under cold conditions, where reaction rates are reduced. Under these circumstances, Balon may pause a substantial fraction of cellular ribosomes that would otherwise be unable to terminate elongation efficiently upon stress. Notably, many phenotypes associated with Stm1 have likewise been observed at low temperatures (Table 1), suggesting that cold stress broadly favors mechanisms that couple elongation arrest to ribosome protection. More generally, distinct stress conditions may require distinct hibernation strategies, and the evolution of bacterial hibernation factors provides insight into how such flexibility is achieved: evolving cells can generate new hibernation factors by fusing HPF to stress response domains and deploy these non-canonical HPF isoforms to initiate stress-specific modes of ribosome hibernation [63].

### 4.2. Distinguishing Hibernating Dimers from Collided Ribosomes

The difficulty in distinguishing blocks in initiation or early elongation from hibernating monosomes is mirrored at the level of two ribosomes: bulk assays similarly struggle to discriminate hibernating ribosomal dimers from ribosome collisions.

Ribosome collisions can be detected by RNase I digestion followed by polysome profiling. In standard ribosome footprinting approaches, sequencing is typically performed on the monosome peak; in contrast, collided ribosomes persist as RNase-resistant disomes after digestion. Accordingly, a characteristic disome peak is observed, and it has been estimated that approximately 6% of ribosomes in fast-proliferating yeast cells are trapped in such RNase-resistant disomes [64]. In general, mRNAs with higher ribosome density—that is, more ribosomes per unit length—exhibit a higher frequency of ribosome collisions, as increased density promotes stochastic, transient encounters between adjacent ribosomes. In addition, disomes frequently accumulate at stop codons, suggesting that slow translation termination increases local ribosome density and thereby promotes collisions [64]. Likewise, collision-induced disomes form in yeast when a fast-moving ribosome encounters a slow-decoding region [65]. Despite their frequent occurrence, ribosome collisions are unlikely to be detrimental because cells employ mechanisms that mitigate their impact. For example, mRNAs with short coding sequences typically exhibit high ribosome density [66] and are therefore expected to experience frequent collisions, yet these transcripts are often translated more efficiently than longer mRNAs [67]. Notably, short coding sequences show reduced sensitivity to codon optimality-dependent changes in elongation rate [67], suggesting that constrained variation in elongation dynamics limits collision-induced interference. In this context, an apparent insensitivity to codon optimality may function as a protective mechanism that mitigates the consequences of frequent ribosome collisions.

Structural analyses of collision-derived disomes further suggest that ribosome collisions on most mRNAs are transient and insufficient to elicit quality control responses. Cryo-EM analyses of disomes isolated after RNase I digestion revealed a distinctive architecture in which two ribosomes are arranged with their 40S subunits facing one another, reminiscent of prokaryotic 100S dimers [64]. However, in this case the ribosomes are linked by a bent mRNA, rather than forming a stable, mRNA-free dimer. Importantly, these disomes adopt a conformation distinct from ribosome-associated quality control (RQC)-inducing di-ribosomes, which trigger Hel2-dependent mRNA decay. Only a small subset of yeast genes, such as *SDD1* [68], which contains clusters of positively charged amino acids (lysine and arginine), robustly elicit RQC. This indicates that the majority of ribosome collisions are transient and do not lead to mRNA decay. Consistent with this interpretation, similar phenomena have been observed in other organisms. In zebrafish, for example, ribosome collisions trigger mRNA decay on transcripts encoding specific zinc-finger proteins, where tandem arrays of positively charged C2H2 zinc-finger motifs repeatedly expose basic nascent polypeptides to the ribosome exit tunnel [69], blocking the elongation of ribosomes.

Ribosome collision frequency can increase under stress conditions, where such events activate dedicated stress response pathways [70]. Because decoding rates are dynamically modulated by tRNA availability, fluctuations in tRNA pools can further influence the likelihood of ribosome collisions [71].

Thus, although disomes can arise from ribosome collisions, these assemblies are fundamentally distinct from vacant ribosomal dimers, which lack an mRNA linker. The two processes may nevertheless overlap, a possibility that remains to be determined.

### 4.3. Quantifying Post-Transcriptional Processes During Stress

Dormant ribosomes, whether monosomes or disomes, arise under nutrient depletion or other stress conditions. Such conditions complicate the quantification of post-transcriptional processes associated with translation, particularly translation rates and mRNA degradation, which are commonly measured to characterize these states. Because each experimental approach introduces its own perturbations, stress conditions can amplify these effects, creating an adverse synergy between the measurement method and the physiological state [72,73]. This interplay is well illustrated under heat stress, where nucleotide analogs used to determine mRNA half-lives can themselves alter decay rates [15]. Accurate quantification therefore requires careful calibration of nucleotide concentration to minimize such effects. Ultimately, the development of new approaches will be required to reliably determine rates of processes such as mRNA degradation in monosomal and polysomal fractions. An important step in this direction is the direct measurement of nascent mRNA synthesis and translation rates [74].

A second major source of methodological bias arises from the widely used polysome profiling procedure. For example, cycloheximide (CHX) is routinely added to stabilize ribosome–mRNA complexes and prevent ribosomal runoff [75]. However, when CHX is dissolved in ethanol, it induces transcription of ribosome biogenesis (RiBi) genes. Further, ethanol alone has a similar, albeit weaker, effect [76]. This can lead to two types of misinterpretation: (i) that RiBi is transcriptionally induced under the tested condition and (ii) that translation efficiency is reduced, because RiBi mRNAs accumulate but are not translated [76]. These artifacts can be mitigated by dissolving CHX in DMSO [76], and by reducing the time between CHX addition and cell harvesting since perturbations display a delayed kinetics [15].

Importantly, ethanol can also accumulate endogenously through glucose fermentation once yeast cells exit the mid-log phase [77]. Thus, part of the translational repression attributed to starvation or stationary-phase physiology may arise indirectly through ethanol production. This further implies that cultures at different starvation or stationary-state stages may behave differently. A related phenomenon has been reported in bacteria, where translation first declines but then appears to re-activate after extended stationary phase (around day 6), as indicated by reassociation of translation factors with ribosomes [78].

In addition to CHX and ethanol, condition-specific accumulation of RiBi mRNAs has been observed with other translation inhibitors, suggesting that this response may reflect a broader feedback mechanism triggered by global translation inhibition [79]. CHX has also been shown to inhibit translation of different mRNAs to different extents in mammalian cells [80], underscoring that inhibition profiles are transcript-specific and context-dependent.

### 4.4. Stress-Induced Changes in Cytoplasmic Biophysical Properties

Several biophysical properties of the cytoplasm change during stress. When polysomes dissociate, the released mRNAs frequently condense with RNA-binding proteins to form stress granules. This reorganization alters the material state of the cytoplasm: both the loss of polysomal scaffolding and the redistribution of mRNAs within the cytosol modify the elastic confinement that normally limits mesoscale diffusion. As a result, cytoplasmic fluidity transiently increases—typically for one to two hours—effectively doubling the diffusion coefficient [81]. While enzymatic activities may decrease during stress, the concomitant increase in diffusivity can accelerate reaction kinetics, potentially masking or counterbalancing the inhibitory effects. These dynamic changes must be considered when interpreting reaction rate constants under stress conditions.

It is also important to note that dormant ribosomes protect the entire ribosome during stress, whereas stress granules primarily safeguard mRNAs together with the small ribosomal subunit. However, recent findings indicate that these compartments are not as distinct as once thought: cryogenic correlative light and electron microscopy has revealed that stress granules can contain substantial amounts of 80S ribosomes [82], suggesting a closer functional and structural interplay between ribosome protection and mRNA sequestration than previously appreciated. These 80S ribosomes are idle, in agreement with the observation that mRNAs must be released from stalled ribosomes to be recruited into stress granules [83].

## 5. Biological and Biotechnological Relevance of Dormant and Inactive Ribosomes

The primary physiological role of ribosome hibernation is presumed to be the protection and preservation of functional ribosomes, thereby enabling rapid reactivation of translation once favorable growth conditions return. Beyond this fundamental function, insights into ribosome hibernation are increasingly being explored for biotechnological and medical applications. For example, cell-free translation systems can be improved by eliminating hibernation factors. In Bacillus subtilis, wild-type extracts display a prominent 100S dimer peak, which disappears upon *hpf* deletion, and translation efficiency increases approximately fourfold under optimal magnesium concentrations [84]. Likewise, in *Saccharomyces cerevisiae*, *STM1* deletion enhances translation yields by roughly twofold [84].

Further, an emerging area concerns the role of silent ribosomes in antibiotic tolerance and resistance, where ribosome hibernation may contribute to cellular persistence during exposure to antibiotics [11,85]. Related mechanisms may also operate in dormant cancer cells, though the involvement of ribosome hibernation in oncology is only beginning to be investigated [86]. Therefore, sensitive detection of silent ribosomes and a clearer understanding of their function in these contexts are likely to provide important insights.

## 6. Summary and Outlook

In prokaryotes, vacant ribosome dimers stabilized by ribosome modulation and hibernation-promoting factors constitute the dominant mechanism for ribosome preservation under stress. In contrast, eukaryotes rely predominantly on dormant monosomes. An interesting evolutionary intermediate is represented by chloroplasts: although the relevant protein can dimerize prokaryotic ribosomes, it fails to induce ribosome dimerization in chloroplasts. This observation suggests that eukaryotic ribosomes are intrinsically less prone to stable dimerization. However, this distinction is not absolute, as collision-derived disomes on mRNAs often resemble ribosome dimers formed through interactions between the small subunits.

In yeast, the principal ribosome-associated preservation factors are Stm1 and Lso2, which protect ribosomes during long-term starvation and facilitate translation restart upon nutrient replenishment, respectively. Remarkably, this mechanism is highly conserved, as evidenced by the mammalian homologs SERBP1 and CCDC124. Nonetheless, vacant 80S ribosomes also form rapidly under short-term stress, even in the absence of Stm1. This raises the possibility that additional ribosome-protective or dimerization factors remain to be identified, or alternatively, that ribosomal subunits possess an inherent tendency to associate weakly in the absence of mRNA under physiological salt conditions—a tendency that is suppressed at higher ionic strength. Together, these observations suggest that formation of vacant ribosomes represents an economical cellular response to stress: while transcription is rapidly downregulated and mRNAs are degraded, ribosomes—being comparatively stable—are preserved and can be rapidly redeployed once favorable conditions return.

Awareness of silent ribosomes relevant for interpretation of data even under moderate stress conditions, as translational output may not scale proportionally with total ribosome abundance—an assumption often made in quantitative studies. It also remains unclear what specific advantages silent ribosomes confer over elongation arrest. Polysomes stalled during elongation are fully compatible with short-term storage and transport of translation-ready mRNAs, particularly in neurons. Vacant monosomes, however, may be preferentially formed under stress, potentially because they permit fluidization of the cytoplasm or confer other, as yet unidentified, advantages. Finally, it remains to be determined whether known or as-yet-unidentified proteins associated with vacant ribosomes contribute to human disease or pathological stress responses.

## Figures and Tables

**Figure 2 ijms-27-00308-f002:**
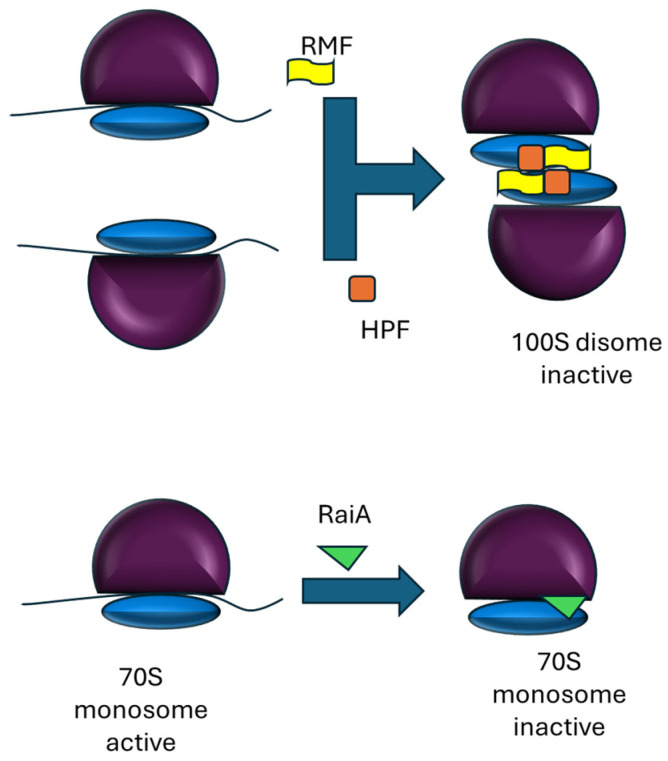
Main routes to the formation of hibernating ribosomes in bacteria. The active monosome (70S) consist of an mRNA and the small (30S, blue) and large (50S, magenta) subunits of the ribosome. The formation of vacant dimeric ribosomes requires RMF and HPF, whereas the formation of inactive monosome is facilitated by RaiA (or YfiA).

**Figure 3 ijms-27-00308-f003:**
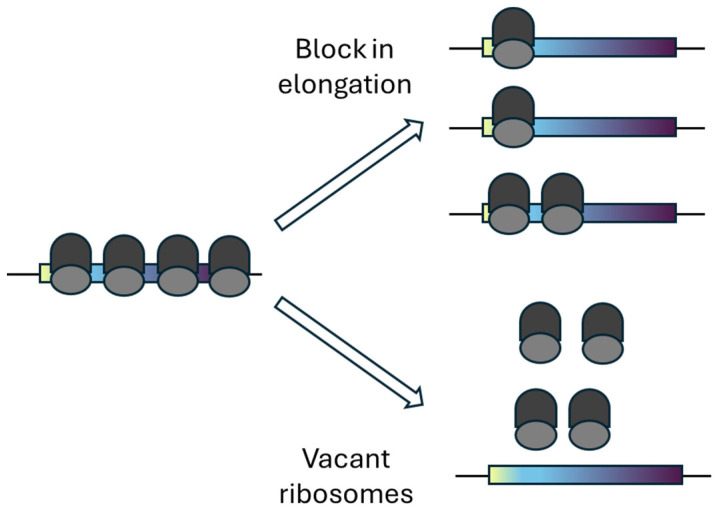
Alternative mechanisms leading to an increase in the monosome peak in the polysome profile. The dark and light gray areas denote the large and small subunits of the ribosome, respectively. A stress-induced collapse of polysomes can cause ribosomes to stall near the start codon (yellow region in the rectangles denoting the mRNAs), giving rise to mRNA-associated monosomes and, to a lesser extent, dimers (**top** panel). Alternatively, stress may lead to the formation of vacant 80S ribosomes (monosomes) in eukaryotes (**bottom** panel).

## Data Availability

No new data were created or analyzed in this study. Data sharing is not applicable to this article.

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
