# Peer review of "Detection and Characterization of the Eukaryotic Vacant Ribosome"

_ijms, 2025, doi:10.3390/ijms27010308_

Round 1

Reviewer 1 Report

Comments and Suggestions for Authors

For many years, translation research has focused on mRNAs and their regulatory transitions between active translation and stress-induced storage. In this review, the authors aim not only to describe the behavior of ribosomes outside active translation but also to discuss the methodological challenges that have hindered the study of dormant ribosomes in eukaryotes. Unlike prokaryotic systems, in which 100S dimers provide clear biochemical signatures, eukaryotic ribosomes lack such obvious markers. This topic is undoubtedly important and fully deserves attention in the pages of this journal.

However, the manuscript in its current form requires some revision. While it is clear that the discussion of ribosome dormancy in eukaryotes is inseparable from the description of the organization of ribosome dormancy in bacteria, by the end of the manuscript, especially in the conclusion, which outlines future prospects, the eukaryotic context completely disappears. In my opinion, it would be helpful to summarize (perhaps in table form) the specific challenges researchers face when studying ribosome dormancy in eukaryotes.

The title itself is problematic, as it conflates two fundamentally different levels of research. "Detection" and "function" are words with different levels of semantic nature. "Detection" refers to a methodological problem, while "function" addresses a biological question. Using these two words in the title creates the impression that the manuscript lacks a clear conceptual focus, making it unclear what the authors' focus is. The title should be changed.

The terminology related to inactive ribosomes must be clearly defined (vacant, dormant, idle, empty, inactive etc.), perhaps, with a distinction between prokaryotic and eukaryotic systems.

It would be beneficial to include a schematic diagram in Section 4 to visually represent the methods, along with an explanation of the key methodological approaches and analysis logic.

There is still uncertainty regarding whether eukaryotic organisms possess homologues (structural or functional) for bacterial RMF/HPF proteins.

The text suggests that PSRP1 has the potential to induce 100S-like dimer formation in chloroplasts, but it also notes that these particles have not been observed in vivo or in vitro - this is a contradiction.

The manuscript states that microorganisms adjust their ribosome content by reducing it through autophagy and other degradation pathways (lines 43-44). However, a question arises: is autophagy specific to microorganisms (whatever they could be), or does it also occur in other organisms?

Other minor corrections:

Species names should be italicized (Saccharomyces cerevisiae, Escherichia coli, Drosophila melanogaster, etc.).

It is not always clear whether the text is referring to genes or proteins (e.g., RMF, HPF, STM1, SERBP1). Gene names are often italicized to distinguish them from their protein products.

In Figure 1's legend, genes (rmf, hpf, and yfiA) are mentioned before being introduced in the text, disrupting the logical flow.

There are inconsistencies and incorrect references to figures in the text. In line 66, the figure reference should be Figure 1 instead of Figure 2, and the paragraph from lines 169-172 refers to Figure 3.

The legend to Figure 2 should include definitions for all labels and colour-coding explanations.

Line 125 has an extra "l" before "less."

Please, spell out abbreviated protein/gene names (for example, "RaiA" and "YfiA").

In line 183, there is an incorrect reference given - it should be either [26] or [27]. Additionally, the year of the study needs to be corrected.

Author Response

REVIEW 1

  1. For many years, translation research has focused on mRNAs and their regulatory transitions between active translation and stress-induced storage. In this review, the authors aim not only to describe the behavior of ribosomes outside active translation but also to discuss the methodological challenges that have hindered the study of dormant ribosomes in eukaryotes. Unlike prokaryotic systems, in which 100S dimers provide clear biochemical signatures, eukaryotic ribosomes lack such obvious markers. This topic is undoubtedly important and fully deserves attention in the pages of this journal.

We thank the reviewer for the insightful comments and suggestions, which we addressed in detail below.

  1. However, the manuscript in its current form requires some revision. While it is clear that the discussion of ribosome dormancy in eukaryotes is inseparable from the description of the organization of ribosome dormancy in bacteria, by the end of the manuscript, especially in the conclusion, which outlines future prospects, the eukaryotic context completely disappears. In my opinion, it would be helpful to summarize (perhaps in table form) the specific challenges researchers face when studying ribosome dormancy in eukaryotes.

We now inserted a new section (“Summary and outlook”) to summarize the conclusions. Furthermore, we added a new section  3.4.        “Neurons: active monosomes and dormant polysomes?” to remind the readers that polysomes can be dormant in some cell types, such as neurons.

  1. The title itself is problematic, as it conflates two fundamentally different levels of research. "Detection" and "function" are words with different levels of semantic nature. "Detection" refers to a methodological problem, while "function" addresses a biological question. Using these two words in the title creates the impression that the manuscript lacks a clear conceptual focus, making it unclear what the authors' focus is. The title should be changed.

We agree, that the current review focuses mostly on detection. Therefore, we changed the title to “Detection and Characterization of the Eukaryotic Vacant Ribosome”.

  1. The terminology related to inactive ribosomes must be clearly defined (vacant, dormant, idle, empty, inactive etc.), perhaps, with a distinction between prokaryotic and eukaryotic systems.

Since different authors use different terminologies, it is relatively difficult to provide uniform definitions. Therefore, we simply stated: "For the purposes of this review, we will treat these terms as interchangeable."

  1. It would be beneficial to include a schematic diagram in Section 4 to visually represent the methods, along with an explanation of the key methodological approaches and analysis logic.

We inserted the Table 3 to summarize the methods.

  1. There is still uncertainty regarding whether eukaryotic organisms possess homologues (structural or functional) for bacterial RMF/HPF proteins.

See answer at the remark regarding the chloroplast.

  1. The text suggests that PSRP1 has the potential to induce 100S-like dimer formation in chloroplasts, but it also notes that these particles have not been observed in vivo or in vitro - this is a contradiction.

We have now expanded the description to resolve this contradiction: “Interestingly, PSRP1 induces 100S-like dimers with purified bacterial ribosomes in vitro, but not with purified chloroplast ribosomes. Furthermore, endogenous chloroplast 100S particles have not been detected”

  1. The manuscript states that microorganisms adjust their ribosome content by reducing it through autophagy and other degradation pathways (lines 43-44). However, a question arises: is autophagy specific to microorganisms (whatever they could be), or does it also occur in other organisms?

We extended the description: “When cells enter stationary phase, growth slows or ceases and the demand for protein synthesis drops. In response, bacteria reduce ribosome abundance through active degradation [3], while eukaryotes rely on multiple turnover pathways, most prominently selective autophagy of ribosomes (ribophagy) [4].”

Other minor corrections:

Species names should be italicized (Saccharomyces cerevisiae, Escherichia coliDrosophila melanogaster, etc.).

We have italicized the species names.

It is not always clear whether the text is referring to genes or proteins (e.g., RMF, HPF, STM1, SERBP1). Gene names are often italicized to distinguish them from their protein products.

We have italicized the gene names.

In Figure 1's legend, genes (rmf, hpf, and yfiA) are mentioned before being introduced in the text, disrupting the logical flow.

We inserted “hibernation factors” before the names to indicate the roles of the specified protein.

There are inconsistencies and incorrect references to figures in the text. In line 66, the figure reference should be Figure 1 instead of Figure 2, and the paragraph from lines 169-172 refers to Figure 3.

We corrected the mistake in line 66 and cited the Fig. 3 for a second time in the respective lines.

The legend to Figure 2 should include definitions for all labels and colour-coding explanations.

The colors are now explained in the figure legend.

Line 125 has an extra "l" before "less."

Removed.

Please, spell out abbreviated protein/gene names (for example, "RaiA" and "YfiA").

Full name is spelled out.

In line 183, there is an incorrect reference given - it should be either [26] or [27]. Additionally, the year of the study needs to be corrected.

The previous reference number [25] was correct. It is mentioned twice, once to describe the findings in the article and a second time to indicate the hypothesis the authors formulated.  Nevertheless, the text was formulated as if it was two different studies. Therefore, we rewrote this section to address both the relevant functional and structural studies:

“The identification of ribosome preservation mechanisms in budding yeast Saccharomyces cerevisiae began with the characterization of Stm1, a protein that binds independently of mRNA to ribosomes and supports cell viability during prolonged starvation [28]. Structural studies of ribosomes purified from cells subjected to brief glucose starvation showed that Stm1 is associated with the 40S subunit, where it blocks the mRNA entry channel [29]. Owing to its translation-inhibitory effect (Table 1), Stm1 was proposed to preserve ribosomes in an inactive state during nutrient limitation, thereby serving as a functional analog of the stress-induced ribosome preservation factors found in bacteria and chloroplasts [29].”

Reviewer 2 Report

Comments and Suggestions for Authors

< !--StartFragment -->

Many organisms defend themselves against environmental stress by inducing ribosome hibernation through hibernation factors to conserve energy. In this manuscript, Delaney and Becskei provide a comprehensive summary of key translation-inhibition mechanisms(and factor involved) in prokaryotes and plant chloroplasts, and discuss the features that facilitated their early characterization, in contrast to the factors that obscured or delayed the recognition of silent ribosomes in eukaryotes. Overall, I found this manuscript to be clearly written, thoughtfully organized, and highly informative. The following comments may help further improve the manuscript:

  1. It is appears that the authors have completely overlooked the identification and characterization of the well conserved eukaryotic hibernation factor LSO2/CCDC124(PMID: 30208026 and 32687489). I would recommend the author to also briefly summarize these work and discuss how the findings may inform the discovery of additional eukaryotic hibernation factors.
  2. In a more recent study, a new bacterial ribosome hibernation mechanism and its responsible protein factor, Balon, were identified and characterized (PMID: 38355796). The authors also performed an evolutionary analysis of this factor in that work and in a follow-up study (doi: 10.1101/2025.11.04.686544). Could the authors comment on whether similar evolutionary approaches might aid in identifying new hibernation factors in eukaryotic organisms.
  3. Could the authors also discuss additional approaches that might facilitate the identification of eukaryotic ribosome hibernation factors

< !--EndFragment -->

Author Response

Many organisms defend themselves against environmental stress by inducing ribosome hibernation through hibernation factors to conserve energy. In this manuscript, Delaney and Becskei provide a comprehensive summary of key translation-inhibition mechanisms(and factor involved) in prokaryotes and plant chloroplasts, and discuss the features that facilitated their early characterization, in contrast to the factors that obscured or delayed the recognition of silent ribosomes in eukaryotes. Overall, I found this manuscript to be clearly written, thoughtfully organized, and highly informative. The following comments may help further improve the manuscript:

We thank the reviewer for the insightful comments and suggestions, which we addressed in detail below.

  1. It is appears that the authors have completely overlooked the identification and characterization of the well conserved eukaryotic hibernation factor LSO2/CCDC124(PMID: 30208026 and 32687489). I would recommend the author to also briefly summarize these work and discuss how the findings may inform the discovery of additional eukaryotic hibernation factors.

We have now summarized also the discovery of the Lso2/ CCDC124  and prepared a new Table (see Table 2 and the accompanying text).

  1. In a more recent study, a new bacterial ribosome hibernation mechanism and its responsible protein factor, Balon, were identified and characterized (PMID: 38355796). The authors also performed an evolutionary analysis of this factor in that work and in a follow-up study (doi: 10.1101/2025.11.04.686544). Could the authors comment on whether similar evolutionary approaches might aid in identifying new hibernation factors in eukaryotic organisms.

Indeed, Balon is a very useful example to show that elongation blockade and hibernation may not be separated at all in some cases. We describe now this phenomenon in the section entitled  “4.1. Distinguishing vacant ribosomes from elongation blockade”.

  1. Could the authors also discuss additional approaches that might facilitate the identification of eukaryotic ribosome hibernation factors

Given that fact, that elongation blockade and hibernation overlap in the case of Balon, there may be a similar overlap between dimers and collided disomes . Therefore, we created a specific subsection on this topic “ Distinction of hibernating dimers from collided ribosomes”, extending the previous the text with more examples and details.

Round 2

Reviewer 1 Report

Comments and Suggestions for Authors

The manuscript has gained clarity and focus in its current version. However, some minor issues related to formatting remain.

Please, check all gene and protein names throughout the manuscript and format them according to the appropriate nomenclature rules. Note that nomenclature in some cases may differ between bacteria and yeast, particularly with respect to capitalization, italics, and the placement of the delta symbol.

For yeasts (in particular S. cerevisiae, gene names should be uppercase and italicized (e.g., STM1); but deletion mutants should be written in lowercase italics (stm1 deletion), with the delta symbol placed after the gene name (e.g., stm1Δ). Protein names should be non-italicized (e.g., Stm1). Please, check the uniformity of protein names. 

For bacteria (e.g., E. coli), gene names should be lowercase and italicized (e.g., rmf, hpf). Place the delta symbol before the gene name (e.g., Δrmf, Δhpf).

These issues affect multiple instances, including lines 118, 119, 204, 206, 210, 211, 214, 244, 563, and Table 1 (e.g., “STM1 deletion” should be written as stm1 deletion or stm1Δ), and others.

Line 65 - it would be helpful to add the word “genes” in the phrase “the hibernation factors rmf, hpf, and yfiA are deleted.”

Line 172 - C. elegans should be italicized.

Tables 1 and 2 - In the column header “First, Last”, please replace the comma with a slash to match the formatting used in the table entries (e.g., Balagopal/Parker).

Author Response

The manuscript has gained clarity and focus in its current version. However, some minor issues related to formatting remain.

We are pleased that the referee found that our major revision improved the focus of the article. The detailed responses to the referee’s minor comments are given below.  

  1. Please, check all gene and protein names throughout the manuscript and format them according to the appropriate nomenclature rules. Note that nomenclature in some cases may differ between bacteria and yeast, particularly with respect to capitalization, italics, and the placement of the delta symbol.

For yeasts (in particular S. cerevisiae, gene names should be uppercase and italicized (e.g., STM1); but deletion mutants should be written in lowercase italics (stm1 deletion), with the delta symbol placed after the gene name (e.g., stm1Δ). Protein names should be non-italicized (e.g., Stm1). Please, check the uniformity of protein names. 

For bacteria (e.g., E. coli), gene names should be lowercase and italicized (e.g., rmf, hpf). Place the delta symbol before the gene name (e.g., Δrmf, Δhpf).

These issues affect multiple instances, including lines 118, 119, 204, 206, 210, 211, 214, 244, 563, and Table 1 (e.g., “STM1 deletion” should be written as stm1 deletion or stm1Δ), and others.

We agree that delta symbol is typically placed before bacterial genes but after the gene names for yeast. We have now followed this rule, reformatting the names accordingly.

However, the deletion of yeast genes is denoted with lower case only when the end-result is shown (e.g. stm1Δ), meaning stm1-deleted cells, i.e. cells harboring a fully deleted gene. However, the deletion of STM1 or STM1 deletion refers to the process, i.e. starting with the wild-type genes, when the gene name is capitalized. Therefore, we kept all these instances in uppercase italics. This nomenclature is followed also  for example in the following article on STM1:

https://pmc.ncbi.nlm.nih.gov/articles/PMC2621192/

“If Stm1 functions to increase Dhh1 activity in some manner, this would predict that a deletion of STM1 reduces the function of Dhh1.”

“To test this possibility, we examined the formation of P-bodies in stm1Δ strains and their ability to accumulate a GFP-tagged version of Dhh1.”

Thus, the  STM1 gene is upper case in the first sentence despite it is mentioned before “deletion”.

On the other hand, we changed the  “deletion of RMF and HPF” to lowercase since bacterial genes are denoted in lowercase independently of whether they are wild-type or mutated. 

  1. Line 65 - it would be helpful to add the word “genes” in the phrase “the hibernation factors rmf, hpf, and yfiA are deleted.”

“the genes of” is inserted ahead of  “the hibernation factors”.

  1. Line 172 - elegans should be italicized.

Done as suggested.

  1. Tables 1 and 2 - In the column header “First, Last”, please replace the comma with a slash to match the formatting used in the table entries (e.g., Balagopal/Parker).

Done as suggested.